# Automatic Rule Extraction from Long Short Term Memory Networks

**W. James Murdoch** [*]
Department of Statistics
UC Berkeley
Berkeley, CA 94709, USA
jmurdoch@berkeley.edu

**Arthur Szlam**
Facebook AI Research
New York City, NY, 10003
aszlam@fb.com

## Abstract

Although deep learning models have proven effective at solving problems in natural language processing, the mechanism by which they come to their conclusions is often unclear. As a result, these models are generally treated as black boxes, yielding no insight of the underlying learned patterns. In this paper we consider Long Short Term Memory networks (LSTMs) and demonstrate a new approach for tracking the importance of a given input to the LSTM for a given output. By identifying consistently important patterns of words, we are able to distill state of the art LSTMs on sentiment analysis and question answering into a set of representative phrases. This representation is then quantitatively validated by using the extracted phrases to construct a simple, rule-based classifier which approximates the output of the LSTM.

## 1 Introduction

Neural network language models, especially recurrent neural networks (RNN), are now standard tools for natural language processing. Amongst other things, they are used for translation Sutskever et al. (2014), language modelling Jozefowicz et al. (2016), and question answering Hewlett et al. (2016). In particular, the Long Short Term Memory (LSTM) Hochreiter & Schmidhuber (1997) architecture has become a basic building block of neural NLP. Although LSTM's are regularly used in state of the art systems, their operation is not well understood. Besides the basic desire from a scientific viewpoint to clarify their workings, it is often the case that it is important to understand why a machine learning algorithm made a particular choice. Moreover, LSTM's are computationally intensive compared to discrete models with lookup tables and pattern matching.

In this work, we describe a novel method for visualizing the importance of specific inputs for determining the output of an LSTM. We then demonstrate that, by searching for phrases which are consistently important, the importance scores can be used to extract simple phrase patterns consisting of one to five words from a trained LSTM. The phrase extraction is first done in a general document classification framework on two different sentiment analysis datasets. We then demonstrate that it can also be specialized to more complex models by applying it to WikiMovies, a recently introduced question answer dataset. To concretely validate the extracted patterns, we use them as input to a rules-based classifier which approximates the performance of the original LSTM.

## 2 Related Work

There are two lines of related work on visualizing LSTMs. First, Hendrik et al. (2016) and Karpathy et al. (2016) analyse the movement of the raw gate activations over a sequence. Karpathy et al. (2016) is able to identify co-ordinates of $c_t$ that correspond to semantically meaningful attributes such as whether the text is in quotes and how far along the sentence a word is. However, most of the cell co-ordinates are harder to interpret, and in particular, it is often not obvious from their activations which inputs are important for specific outputs.

---

[*]Work started during an internship at Facebook AI Research

Another approach that has emerged in the literature Alikaniotis et al. (2016) Denil et al. (2015) Bansal et al. (2016) is for each word in the document, looking at the norm of the derivative of the loss function with respect to the embedding parameters for that word. This bridges the gap between high-dimensional cell state and low-dimensional outputs. These techniques are general- they are applicable to visualizing the importance of sets of input coordinates to output coordinates of any differentiable function. In this work, we describe techniques that are designed around the structure of LSTM's, and show that they can give better results in that setting.

A recent line of work Li et al. (2016) Hewlett et al. (2016) Rajpurkar et al. (2016) Miller et al. (2016) has focused on neural network techniques for extracting answers directly from documents. Previous work had focused on Knowledge Bases (KBs), and techniques to map questions to logical forms suitable for querying them. Although they are effective within their domain, KBs are inevitably incomplete, and are thus an unsatisfactory solution to the general problem of question-answering. Wikipedia, in contrast, has enough information to answer a far broader array of questions, but is not as easy to query. Originally introduced in Miller et al. (2016), the WikiMovies dataset consists of questions about movies paired with Wikipedia articles.

## 3 WORD IMPORTANCE SCORES IN LSTMS

We present a novel decomposition of the output of an LSTM into a product of factors, where each term in the product can be interpreted as the contribution of a particular word. Thus, we can assign importance scores to words according to their contribution to the LSTM's prediction

### 3.1 LONG SHORT TERM MEMORY NETWORKS

Over the past few years, LSTMs have become an important part of neural NLP systems. Given a sequence of word embeddings $x_1, ..., x_T \in \mathbb{R}^d$, an LSTM processes one word at a time, keeping track of cell and state vectors $(c_1, h_1), ..., (c_T, h_T)$ which contain information in the sentence up to word $i$. $h_t$ and $c_t$ are computed as a function of $x_t, c_{t-1}$ using the below updates

$$f_t = \sigma(W_f x_t + V_f h_{t-1} + b_f) \tag{1}$$
$$i_t = \sigma(W_i x_t + V_i h_{t-1} + b_i) \tag{2}$$
$$o_t = \sigma(W_o x_t + V_o h_{t-1} + b_o) \tag{3}$$
$$\tilde{c}_t = \tanh(W_c x_t + V_c h_{t-1} + b_c) \tag{4}$$
$$c_t = f_t c_{t-1} + i_t \tilde{c}_t \tag{5}$$
$$h_t = o_t \odot \tanh(c_t) \tag{6}$$

As initial values, we define $c_0 = h_0 = 0$. After processing the full sequence, a probability distribution over $C$ classes is specified by $p$, with

$$p_i = \text{SoftMax}(W h_T) = \frac{e^{W_i h_T}}{\sum_{j=1}^{C} e^{W_j h_t}} \tag{7}$$

where $W_i$ is the $i$'th row of the matrix $W$

### 3.2 DECOMPOSING THE OUTPUT OF A LSTM

We now show that we can decompose the numerator of $p_i$ in Equation 7 into a product of factors, and interpret those factors as the contribution of individual words to the predicted probability of class $i$. Define

$$\beta_{i,j} = \exp\left(W_i(o_T \odot (\tanh(c_j) - \tanh(c_{j-1})))\right), \tag{8}$$

so that

$$\exp(W_i h_T) = \exp\left(\sum_{j=1}^{T} W_i(o_T \odot (\tanh(c_j) - \tanh(c_{j-1})))\right) = \prod_{j=1}^{T} \beta_{i,j}.$$

As $\tanh(c_j) - \tanh(c_{j-1})$ can be viewed as the update resulting from word $j$, so $\beta_{i,j}$ can be interpreted as the multiplicative contribution to $p_i$ by word $j$.

### 3.3 AN ADDITIVE DECOMPOSITION OF THE LSTM CELL

We will show below that the $\beta_{i,j}$ capture some notion of the importance of a word to the LSTM's output. However, these terms fail to account for how the information contributed by word $j$ is affected by the LSTM's forget gates between words $j$ and $T$. Consequently, we empirically found that the importance scores from this approach often yield a considerable amount of false positives. A more nuanced approach is obtained by considering the additive decomposition of $c_T$ in equation (9), where each term $e_j$ can be interpreted as the contribution to the cell state $c_T$ by word $j$. By iterating the equation $c_t = f_t c_{t-1} + i_t \tilde{c}_t$, we get that

$$c_T = \sum_{i=1}^{T} (\prod_{j=i+1}^{T} f_j) i_i \tilde{c}_i = \sum_{i=1}^{T} e_{i,T} \tag{9}$$

This suggests a natural definition of an alternative score to the $\beta_{i,j}$, corresponding to augmenting the $c_j$ terms with products of forget gates to reflect the upstream changes made to $c_j$ after initially processing word $j$.

$$\exp(W_i h_T) = \prod_{j=1}^{T} \exp\left( W_i(o_T \odot (\tanh(\sum_{k=1}^{j} e_{k,T}) - \tanh(\sum_{k=1}^{j-1} e_{k,T}))) \right) \tag{10}$$

$$= \prod_{j=1}^{T} \exp\left( W_i(o_T \odot (\tanh((\prod_{k=j+1}^{t} f_k)c_j) - \tanh((\prod_{k=j}^{t} f_k)c_{j-1}))) \right) \tag{11}$$

$$= \prod_{j=1}^{T} \gamma_{i,j} \tag{12}$$

## 4 PHRASE EXTRACTION FOR DOCUMENT CLASSIFICATION

We now introduce a technique for using our variable importance scores to extract phrases from a trained LSTM. To do so, we search for phrases which consistently provide a large contribution to the prediction of a particular class relative to other classes. The utility of these patterns is validated by using them as input for a rules based classifier. For simplicity, we focus on the binary classification case.

### 4.1 PHRASE EXTRACTION

A phrase can be reasonably described as predictive if, whenever it occurs, it causes a document to both be labelled as a particular class, and not be labelled as any other. As our importance scores introduced above correspond to the contribution of particular words to class predictions, they can be used to score potential patterns by looking at a pattern's average contribution to the prediction of a given class relative to other classes. More precisely, given a collection of $D$ documents $\{\{x_{i,j}\}_{i=1}^{N_d}\}_{j=1}^{D}$, for a given phrase $w_1, ..., w_k$ we can compute scores $S_1, S_2$ for classes 1 and 2, as well as a combined score $S$ and class $C$ as

$$S_1(w_1, ..., w_k) = \frac{\text{Average}_{j,b} \left\{ \prod_{l=1}^{k} \beta_{1,b+l,j} | x_{b+i,j} = w_i, i = 1, ..., k \right\}}{\text{Average}_{j,b} \left\{ \prod_{l=1}^{k} \beta_{2,b+l,j} | x_{b+i,j} = w_i, i = 1, ..., k \right\}} \tag{13}$$

$$S_2(w_1, .., w_k) = \frac{1}{S_1(w_1, ..., w_k)} \tag{14}$$

$$S(w_1, ..., w_k) = \max_i (S_i(w_1, ..., w_k)) \tag{15}$$

$$C(w_1, ..., w_k) = \text{argmax}_i (S_i(w_1, ..., w_k)) \tag{16}$$

where $\beta_{i,j,k}$ denotes $\beta_{i,j}$ applied to document $k$.

The numerator of $S_1$ denotes the average contribution of the phrase to the prediction of class 1 across all occurrences of the phrase. The denominator denotes the same statistic, but for class 2. Thus, if

$S_1$ is high, then $w_1, ..., w_k$ is a strong signal for class 1, and likewise for $S_2$. We propose to use $S$ as a score function in order to search for high scoring, representative, phrases which provide insight into the trained LSTM, and $C$ to denote the class corresponding to a phrase.

In practice, the number of phrases is too large to feasibly compute the score of them all. Thus, we approximate a brute force search through a two step procedure. First, we construct a list of candidate phrases by searching for strings of consecutive words $j$ with importance scores $\beta_{i,j} > c$ for any $i$ and some threshold $c$; in the experiments below we use $c = 1.1$. Then, we score and rank the set of candidate phrases, which is much smaller than the set of all phrases.

## 4.2 Rules based classifier

The extracted patterns from Section 4.1 can be used to construct a simple, rules-based classifier which approximates the output of the original LSTM. Given a document and a list of patterns sorted by descending score given by $S$, the classifier sequentially searches for each pattern within the document using simple string matching. Once it finds a pattern, the classifier returns the associated class given by $C$, ignoring the lower ranked patterns. The resulting classifier is interpretable, and despite its simplicity, retains much of the accuracy of the LSTM used to build it.

## 5 Experiments

We now present the results of our experiments.

## 5.1 Training Details

We implemented all models in Torch using default hyperparameters for weight initializations. For WikiMovies, all documents and questions were pre-processed so that multiple word entities were concatenated into a single word. For a given question, relevant articles were found by first extracting from the question the rarest entity, then returning a list of Wikipedia articles containing any of those words. We use the pre-defined splits into train, validation and test sets, containing 96k, 10k and 10k questions, respectively. The word and hidden representations of the LSTM were both set to dimension 200 for WikiMovies, 300 and 512 for Yelp, and 300 and 150 for Stanford Sentiment Treebank. All models were optimized using Adam Kingma & Ba (2015) with the default learning rate of 0.001 using early stopping on the validation set. For rule extraction using gradient scores, the product in the reward function is replaced by a sum. In both datasets, we found that normalizing the gradient scores by the largest gradient improved results.

## 5.2 Sentiment Analysis

We first applied the document classification framework to two different sentiment analysis datasets. Originally introduced in Zhang et al. (2015), the Yelp review polarity dataset was obtained from the Yelp Dataset Challenge and has train and test sets of size 560,000 and 38,000. The task is binary prediction for whether the review is positive (four or five stars) or negative (one or two stars). The reviews are relatively long, with an average length of 160.1 words. We also used the binary classification task from the Stanford Sentiment Treebank (SST) Socher et al. (2013), which has less data with train/dev/test sizes of 6920/872/1821, and is done at a sentence level, so has much shorter document lengths.

We report results in Table 1 for seven different models. We report state of the art results from prior work using convolutional neural networks; Kim (2014) for SST and Zhang et al. (2015) for Yelp. We also report our LSTM baselines, which are competitive with state of the art, along with the three different pattern matching models described above. For SST, we also report prior results using bag of words features with Naive Bayes.

The additive cell decomposition pattern equals or outperforms the cell-difference patterns, which handily beat the gradient results. This coincides with our empirical observations regarding the information contained within the importance measures, and validates our introduced measure. The differences between measures become more pronounced in Yelp, as the longer document sizes provide more opportunities for false positives.

| Model | Yelp Polarity | Stanford Sentiment Treebank |
|---|---|---|
| Large word2vec CNN Zhang et al. (2015) | 95.4 | - |
| CNN-multichannel Kim (2014) | - | 88.1 |
| Naive Bayes Socher et al. (2013) | - | 82.6 |
| LSTM | 95.3 | 87.3 |
| Cell Decomposition Pattern Matching | 86.5 | 76.2 |
| Cell-Difference Pattern Matching | 81.2 | 77.4 |
| Gradient Pattern Matching | 65.0 | 68.0 |

Table 1: Test accuracy on sentiment analysis. See section 5.2 for further descriptions of the models.

Although our pattern matching algorithms underperform other methods, we emphasize that pure performance is not our goal, nor would we expect more from such a simple model. Rather, the fact that our method provides reasonable accuracy is one piece of evidence, in addition to the qualitative evidence given later, that our word importance scores and extracted patterns contain useful information for understanding the actions of a LSTM.

## 5.3 WIKIMOVIES

Although document classification comprises a sizeable portion of current research in natural language processing, much recent work focuses on more complex problems and models. In this section, we examine WikiMovies, a recently introduced question answer dataset, and show that with some simple modifications our approach can be adapted to this problem.

### 5.3.1 DATASET

WikiMovies is a dataset consisting of more than 100,000 questions about movies, paired with relevant Wikipedia articles. It was constructed using the pre-existing dataset MovieLens, paired with templates extracted from the SimpleQuestions dataset Bordes et al. (2015), a open-domain question answering dataset based on Freebase. They then selected a set of Wikipedia articles about movies by identifying a set of movies from OMDb that had an associated article by title match, and kept the title and first section for each article.

For a given question, the task is to read through the relevant articles and extract the answer, which is contained somewhere within the text. The dataset also provides a list of 43k entities containing all possible answers.

### 5.3.2 LSTMs FOR WIKIMOVIES

We propose a simplified version of recent work Li et al. (2016). Given a pair of question $x_1^q, ..., x_N^q$ and document $x_1^d, ..., x_T^d$, we first compute an embedding for the question using a LSTM. Then, for each word $t$ in the document, we augment the word embedding $x_t$ with the computed question embedding. This is equivalent to adding an additional term which is linear in the question embedding into the gate equations 3-6, allowing the patterns an LSTM absorbs to be directly conditioned upon the question at hand.

$$h_t^q = \text{LSTM}(x_t^q) \tag{17}$$

$$h_t = \text{LSTM}(x_t^d \| h_N^q) \tag{18}$$

Having run the above model over the document while conditioning on a question, we are given contextual representations $h_1, ..., h_T$ of the words in the document. For each entity $t$ in the document

| Model | Test accuracy |
|---|---|
| KV-MemNN IE | 68.3 |
| KV-MemNN Doc | 76.2 |
| LSTM | 80.1 |
| Cell Decomposition Pattern Matching | 74.3 |
| Cell-Difference Pattern Matching | 69.4 |
| Gradient Pattern Matching | 57.4 |

Table 2: Test results on WikiMovies, measured in % hits@1. See Section 5.3.4 for further descriptions of the models.

we use $p_t$ to conduct a binary prediction for whether or not the entity is the answer. At test time, we return the entity with the highest probability as the answer.

$$p_t = \text{SoftMax}(W h_t) \tag{19}$$

### 5.3.3 PHRASE EXTRACTION

We now introduce some simple modifications that were useful in adapting our pattern extraction framework to this specific task. First, in order to define the set of classifications problems to search over, we treat each entity $t$ within each document as a separate binary classification task with corresponding predictor $p_t$. Given this set of classification problems, rather than search over the space of all possible phrases, we restrict ourselves to those ending at the entity in question. We also distinguish patterns starting at the beginning of the document with those that do not and introduce an entity character into our pattern vocabulary, which can be matched by any entity. Template examples can be seen below, in Table 4. Once we have extracted a list of patterns, in the rules-based classifier we only search for positive examples, and return as the answer the entity matched to the highest ranked positive pattern.

### 5.3.4 RESULTS

We report results on six different models in Tables 2 and 3. We show the results from Miller et al. (2016), which fit a key-value memory network (KV-MemNN) on representations from information extraction (IE) and raw text (Doc). Next, we report the results of the LSTM described in Section 5.3.2. Finally, we show the results of using three variants of the pattern matching algorithm described in Section 5.3.3: using patterns extracted using the additive decomposition (cell decomposition), difference in cells approaches (cell-difference) and gradient importance scores (gradient), as discussed in Section 2. Performance is reported using the accuracy of the top hit over all possible answers (all entities), i.e. the hits@1 metric.

As shown in Table 2, our LSTM model surpasses the prior state of the art by nearly 4%. Moreover, our automatic pattern matching model approximates the LSTM with less than 6% error, which is surprisingly small for such a simple model, and falls within 2% of the prior state of the art. Similarly to sentiment analysis, we observe a clear ordering of the results across question categories, with our cell decomposition scores providing the best performance, followed by the cell difference and gradient scores.

## 6 DISCUSSION

### 6.1 LEARNED PATTERNS

We present extracted patterns for both sentiment tasks, and some WikiMovies question categories in Table 4. These patterns are qualitatively sensible, providing further validation of our approach. The increased size of the Yelp dataset allowed for longer phrases to be extracted relative to SST.

| | KV-MemNN IE | KV-MemNN Doc | LSTM | Cell Decomp RE | Cell Diff RE | Gradient RE |
|---|---|---|---|---|---|---|
| Actor to Movie | 66 | 83 | 82 | 78 | 77 | 78 |
| Director to Movie | 78 | 91 | 84 | 82 | 84 | 83 |
| Writer to Movie | 72 | 91 | 88 | 88 | 89 | 88 |
| Tag to Movie | 35 | 49 | 49 | 38 | 38 | 38 |
| Movie to Year | 75 | 89 | 89 | 84 | 84 | 84 |
| Movie to Writer | 61 | 64 | 86 | 79 | 72 | 63 |
| Movie to Actor | 64 | 64 | 84 | 75 | 73 | 67 |
| Movie to Director | 76 | 79 | 88 | 86 | 85 | 45 |
| Movie to Genre | 84 | 86 | 72 | 65 | 42 | 21 |
| Movie to Votes | 92 | 92 | 67 | 67 | 67 | 67 |
| Movie to Rating | 75 | 92 | 33 | 25 | 25 | 25 |
| Movie to Language | 62 | 84 | 72 | 67 | 66 | 44 |
| Movie to Tags | 47 | 48 | 58 | 44 | 30 | 6 |

Table 3: Results broken down by question category. See section 5.3.4 for further descriptions of the models.

| Category | Top Patterns |
|---|---|
| **Yelp Polarity Positive** | definitely come back again., love love love this place, great food and great service., highly recommended!, will definitely be coming back, overall great experience, love everything about, hidden gem. |
| **Yelp Polarity Negative** | worst customer service ever, horrible horrible horrible, won't be back, disappointed in this place, never go back there, not worth the money, not recommend this place |
| **SST Positive** | riveting documentary, is a real charmer, funny and touching, well worth your time, journey of the heart, emotional wallop, pleasure to watch, the whole family, cast is uniformly superb, comes from the heart, best films of the year, surprisingly funny, deeply satisfying |
| **SST Negative** | pretentious mess ..., plain bad, worst film of the year, disappointingly generic, fart jokes, banal dialogue, poorly executed, waste of time, a weak script, dullard, how bad it is, platitudes, never catches fire, tries too hard to be, bad acting, untalented artistes, derivative horror film, lackluster |
| **WikiMovies movie to writer** | film adaptation of Charles Dickens', film adapted from ENT, by journalist ENT, written by ENT |
| **WikiMovies movie to actor** | western film starring ENT, starring Ben Affleck, . The movie stars ENT, that stars ENT |
| **WikiMovies movie to language** | is a 2014 french, icelandic, finnish, russian, danish, bengali, dutch, original german, zulu,czech, estonian, mandarin, filipino, hungarian |

Table 4: Selected top patterns using cell decomposition scores, ENT denotes an entity placeholder

| Sentiment | Pattern | Sentence |
|---|---|---|
| Negative | gets the job done | Still, it gets the job done — a sleepy afternoon rental |
| Negative | is a great | This is a great subject for a movie, but Hollywood has squandered the opportunity, using is as a prop for a warmed-over melodrama and the kind of choreographed mayhem that director John Woo has built his career on. |
| Negative | happy ending | The story loses its bite in a last-minute happy ending that's even less plausible than the rest of the picture. |
| Negative | witty dialogue | An often-deadly boring, strange reading of a classic whose witty dialogue is treated with a baffling casual approach. |
| Positive | mess | The film is just a big, gorgeous, mind-blowing, breath-taking mess |

Table 5: Examples from Stanford sentiment treebank which are correctly labelled by our LSTM and incorrectly labelled by our rules-based classifier. The matched pattern is highlighted

## 6.2 APPROXIMATION ERROR BETWEEN LSTM AND PATTERN MATCHING

Although our approach is able to extract sensible patterns and achieve reasonable performance, there is still an approximation gap between our algorithm and the LSTM. In Table 5 we present some examples of instances where the LSTM was able to correctly classify a sentence, and our algorithm was not, along with the pattern used by our algorithm. At first glance, the extracted patterns are sensible, as "gets the job done" or "witty dialogue" are phrases you'd expect to see in a positive review of a movie. However, when placed in the broader context of these particular reviews, they cease to be predictive. This demonstrates that, although our work is useful as a first-order approximation, there are still additional relationships that an LSTM is able to learn from data.

## 6.3 COMPARISON BETWEEN WORD IMPORTANCE MEASURES

While the prediction accuracy of our rules-based classifier provides quantitative validation of the relative merits of our visualizations, the qualitative differences are also insightful. In Table 6, we provide a side-by-side comparison between the different measures. As discussed before, the difference in cells technique fails to account for how the updates resulting from word $j$ are affected by the LSTM's forget gates between when the word is initially processed and the answer. Consequently, we empirically found that without the interluding forget gates to dampen cell movements, the variable importance scores were far noisier than in additive cell decomposition approach. Under the additive cell decomposition, it identifies the phrase 'it stars', as well as the actor's name Aqib Khan as being important, a sensible conclusion. Moreover, the vast majority of words are labelled with an importance score of 1, corresponding to irrelevant. On the other hand, the difference in cells approach yields widely changing importance scores, which are challenging to interpret. In terms of noise, the gradient measures seem to lie somewhere in the middle. These patterns are broadly consistent with what we have observed, and provide qualitative validation of our metrics.

## 7 CONCLUSION

In this paper, we introduced a novel method for visualizing the importance of specific inputs in determining the output of an LSTM. By searching for phrases which consistently provide large contributions, we are able to distill trained, state of the art, LSTMs into an ordered set of representative phrases. We quantitatively validate the extracted phrases through their performance in a simple, rules-based classifier. Results are shown in a general document classification framework, then specialized to a more complex, recently introduced, question answer dataset. Our introduced measures provide superior predictive ability and cleaner visualizations relative to prior work. We believe that this represents an exciting new paradigm for analysing the behaviour of LSTM's.

| Additive cell decomposition | Difference in cell values | Gradient |
| --- | --- | --- |
| west is west is a 2010 british comedy - drama film , which is a sequel to the 1999 comedy " east is east " . it stars aqib khan | west is west is 2010 british comedy - drama film , which is a sequel to the 1999 comedy " " it stars aqib khan | west is west is a 2010 british comedy - drama film , which is a sequel to the 1999 comedy " east is east " . it stars aqib khan |

Table 6: Comparison of importance scores acquired by three different approaches, conditioning on the question "the film west is west starred which actors?". Bigger and darker means more important.

ACKNOWLEDGEMENTS

This research was partially funded by Air Force grant FA9550-14-1-0016. It was also supported by the Center for Science of Information (CSoI), an US NSF Science and Technology Center, under grant agreement CCF-0939370.

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

## 8   APPENDIX - HEAT MAPS

We provide an example heat map using the cell decomposition metric for each class in both sentiment analysis datasets, and selected WikiMovie question categories

| Dataset | Category | Heat Map |
|---|---|---|
| Yelp Polarity | Positive | we went here twice for breakfast . had the ba-nanas foster waffles with fresh whipped cream , they were amazing ! ! perfect seat out side on the terrace |
| Yelp Polarity | Negative | call me spoiled ...this sushi is gross and the orange chicken , well it was so thin i don 't think it had chicken in it. go somewhere else |
| Stanford Sentiment | Positive | Whether or not you 're enlightened by any of Derrida 's lectures on " the other " and " the self , " Derrida is an undeniably fascinating and playful fellow |
| Stanford Sentiment | Negative | ... begins with promise , but runs aground after being snared in its own tangled plot |

| Pattern | Question | Heat Map |
|---------|----------|----------|
| Movie to Year | What was the release year of another 48 hours? | another 48 hrs is a 1990 |
| Movie to Writer | Which person wrote the movie last of the dogmen? | last of the dogmen is a 1995 western adventure film written and directed by tab murphy |
| Movie to Actor | Who acted in the movie thunderbolt? | thunderbolt ( ) ( " piklik foh " ) is a 1995 hong kong action film starring jackie chan |
| Movie to Director | Who directed bloody bloody bible camp? | bloody bloody bible cam p is a 2012 american horror - comedy /s platter film . the film was directed by vito trabucco |
| Movie to Genre | What genre is trespass in? | trespass is a 1992 action |
| Movie to Votes | How would people rate the pool? | though filmed in hindi , a language smith didn 't know , the film earned good* |
| Movie to Rating | How popular was les miserables? | les mis rables is a 1935 american drama film starring fredric march and charles laughton based upon the famous |
| Movie to Tags | Describe rough magic? | rough magic is a 1995 comedy film directed by clare peploe and starring bridget fonda , russell crowe |
| Movie to Language | What is the main language in fate? | fate ( ) is a 2001 turkish |

