# Peer review of "Automatic Rule Extraction from Long Short Term Memory Networks"

_ICLR 2017 — accepted_

[Official Review · AnonReviewer2 · rating 7 · confidence 3 · 15 Dec 2016]

This paper proposes a novel method for extracting rule-based classifiers from trained LSTM models. The proposed method is applied to a factoid question-answering task, where it is demonstrated that the extracted rules perform comparatively to the original LSTM. The analysis of the extracted rules illustrate the features the LSTM model picks up on.

Analyzing and visualizing the computations carried out by RNNs in order to understand the functions they compute is an important direction of research. This sort of analysis will help us understand the pitfalls of RNNs, and how we can improve them. Although the approach taken is relatively inflexible - each rule is defined as an ordered sequence of words - the authors experiment with three different scores for picking salient words (state-difference, cell-difference and gradient) and their approach yields comparable performance, which suggests that the extracted rules mimic the RNN closely. The results are also somewhat surprising, since most of the rules consist only of two or three words.

It would have been interesting to try extend the approach on other natural language processing tasks, such as machine translation. Presumably the rules learned here will be quite different.

Other comments:
- Eq. (12) is over-parametrized with two vectors $P$ and $Q$. The same function can be computed with a single vector. This becomes clear when you divide both the numerator and denominator by $e^{P h_t}$.
- Section 4.1. Is it correct that this section is focused on the forward LSTM? If so, please clarify it in the text.
- In Eq. (13), define $c_0 = 0$.
- Eq. (13) is exactly the same as Eq. (15). Is there a mistake?
- In Table 1, third column should have word "film" highlighted.
- "are shown in 2" -> "are shown in Table 2".
- Since there are some problems representing numbers, it may help to replace each digit with the hashtag symbol #.

[Official Review · AnonReviewer1 · rating 7 · confidence 3 · 17 Dec 2016 (modified: 21 Jan 2017)]
**Nice idea to improve understanding of LSTM models**

This work proposes a pattern extraction method to both understand what a trained LSTM has learnt and to allow implementation of a hand-coded algorithm that performs similarly to the LSTM. Good results are shown on one dataset for one model architecture so it is unclear how well this approach will generalize, however, it seems it will be a useful way to understand and debug models.

The questions in WikiMovies seem to be generated from templates and so this pattern matching approach will likely work well. However, from the experiments it's not clear if this will extend to other types of Q&A tasks where the answer may be free form text and not be a substring in the document. Is the model required to produce a continuous span over the original document?

The approach also seems to have some deficiencies in how it handles word types such as numbers or entity names. This can be encoded in the embedding for the word but from the description of the algorithm, it seems that the approach requires an entity detector. Does this mean that the approach is unable to determine when it has reached an entity from the decomposition of the output of the LSTM? The results where 'manual pattern matching' where explicit year annotations are used, seem to show that the automatic method is unable to deal with word types.

It would also be good to see an attention model as a baseline in addition to the gradient-based baseline.

Minor comments:
- P and Q seem to be undefined.
- Some references seem to be bad, e.g. in section 5.1: 'in 1' instead of 'in table 1'. Similarly above section 7: 'as shown in 3' and in section 7.1.
- In the paragraph above section 6.3: 'adam' -> 'Adam'.

[Official Review · AnonReviewer3 · rating 7 · confidence 4 · 19 Dec 2016 (modified: 24 Jan 2017)]

EDIT: the revisions made to this paper are very thorough and address many of my concerns, and the paper is also easier to understand. i recommend the latest version of this paper for acceptance and have increased my score.

This paper presents a way of interpreting LSTM models, which are notable for their opaqueness. In particular, the authors propose decomposing the LSTM's predictions for a QA task into importance scores for words, which are then used to generate patterns that are used to find answers with a simple matching algorithm. On the WikiMovies dataset, the extracted pattern matching method achieves accuracies competitive with a normal LSTM, which shows the power of the proposed approach. 

I really like the motivation of the paper, as interpreting LSTMs is definitely still a work-in-progress, and the high performance of the pattern matching was surprising. However, several details of the pattern extraction process are not very clear, and  the evaluation is conducted on a very specific task, where predictions are made at every word. As such, I recommend the paper in its current form as a weak accept but hope that the authors clarify their approach, as I believe the proposed method is potentially useful for NLP researchers.

Comments:
- Please introduce in more detail the specific QA tasks you are applying your models on before section 3.3, as it's not clear at that point that the answer is an entity within the document.
- 3.3: is the softmax predicting a 0/1 value (e.g., is this word the answer or not?)
- 3.3: what are the P and Q vectors? do you just mean that you are transforming the hidden state into a 2-dimensional vector for binary prediction?
- how does performance of the pattern matching change with different cutoff constant values?
- 5.2: are there questions whose answers are not entities? 
- how could the proposed approach be used when predictions aren't made at every word? is there any extension for, say, sentence-level sentiment classification?

[Author Response · W. James Murdoch · 14 Jan 2017]
**Updated paper**

In response to helpful comments from reviewers, we have just uploaded a revision. The main changes are as follows

- In response to requests for extensions to other datasets, we now have results on 2 different binary sentiment analysis datasets - Stanford Sentiment Treebank and Yelp reviews

- We introduced a simpler, more general approach for extracting rules from LSTMs trained on document classification, and demonstrate that with some easy modifications it can replace our prior, more complex, rule extraction mechanism on WikiMovies.

- Our rules now take the form of simple phrases, rather than allowing for variable-sized gaps between words as before

- Our LSTM baseline on WikiMovies is now SOTA by nearly 4%, and the automatically extracted patterns outperform the manual patterns in the earlier version.

- We added a discussion on instances correctly classified by the LSTM, but incorrectly classified by our rules-based algorithm

- For simplicity, we changed our WikiMovies baseline to a unidirectional LSTM, and removed bidirectional LSTMs from the paper. Extension of our new approach to bidirectional LSTMs would be straightforward, but we feel would add unneeded complexity to the presentation

All told, we feel that these algorithms and results are simpler, more powerful and more general than our prior work, and we look forward to discussing them.

[Public Comment · (anonymous) · 12 Feb 2017]
**A typo in the published version?**

The equation between (8) - (9) seems to be incorrect, as the left hand side of which should be p_i.

[Final Decision · Program Chairs · 06 Feb 2017]
**ICLR committee final decision**

Timely topic (interpretability of neural models for NLP), interesting approach, surprising results.